# Genetic Analysis and Fine Mapping of a New Rice Mutant, Leaf Tip Senescence 2

**DOI:** 10.3390/ijms25137082

**Published:** 2024-06-27

**Authors:** Yongtao Cui, Jian Song, Liqun Tang, Xiaozheng Xu, Xinlu Peng, Honghuan Fan, Jianjun Wang

**Affiliations:** 1Institute of Crops and Nuclear Technology Utilization, Zhejiang Academy of Agricultural Sciences, Hangzhou 310021, China; song521125@163.com (J.S.); liquntang2013@126.com (L.T.); xixi615@163.com (H.F.); 2College of Landscape and Architecture, Zhejiang A&F University, Hangzhou 311300, China; 19550178539@163.com (X.X.); 15057155749@163.com (X.P.)

**Keywords:** rice, wilted leaf tips, senescence, transposon protein

## Abstract

Premature leaf senescence significantly reduces rice yields. Despite identifying numerous factors influencing these processes, the intricate genetic regulatory networks governing leaf senescence demand further exploration. We report the characterization of a stably inherited, ethyl methanesulfonate(EMS)-induced rice mutant with wilted leaf tips from seedling till harvesting, designated *lts2*. This mutant exhibits dwarfism and early senescence at the leaf tips and margins from the seedling stage when compared to the wild type. Furthermore, *lts2* displays a substantial decline in both photosynthetic activity and chlorophyll content. Transmission electron microscopy revealed the presence of numerous osmiophilic granules in chloroplast cells near the senescent leaf tips, indicative of advanced cellular senescence. There was also a significant accumulation of H_2_O_2_, alongside the up-regulation of senescence-associated genes within the leaf tissues. Genetic mapping situated *lts2* between SSR markers Q1 and L12, covering a physical distance of approximately 212 kb in chr.1. No similar genes controlling a premature senescence leaf phenotype have been identified in the region, and subsequent DNA and bulk segregant analysis (BSA) sequencing analyses only identified a single nucleotide substitution (C-T) in the exon of LOC_Os01g35860. These findings position the *lts2* mutant as a valuable genetic model for elucidating chlorophyll metabolism and for further functional analysis of the gene in rice.

## 1. Introduction

Given its status as an essential model crop and staple cereal in China, the enhancement of rice yield assumes critical importance. Early leaf senescence, typically impacting rice yield [1], warrants keen scientific attention. Consequently, the cloning and elucidation of the genetic regulatory mechanisms associated with premature leaf senescence in rice has the potential to provide a theoretical basis for improving the photosynthetic efficiency of rice during its late reproductive stages, which could contribute to maintaining high rice yields. Befitting its complexities, leaf senescence involves intricate genetic regulatory networks encompassing a cadre of genes, jointly referred to as senescence-associated genes (SAGs). A gamut of developmental processes has been implicated in leaf senescence, as illuminated by studies on Arabidopsis, rice and other plants [1,2,3,4,5,6]. For instance, research on Arabidopsis has revealed that genes residing on the ubiquitylation pathway, such as *ORE9* and *APG7*, mediate senescence through the recycling of nutrients [7,8]. Beneath dark-induced conditions, mutants of the *CPR5/HYS1* gene, known to be involved in cell proliferation and growth, exhibit a premature senescence phenotype [9]. Conversely, the *OLD1* gene, associated with senescence, serves to maintain the redox equilibrium within the cellular milieu [10]. In rice, genes linked to the developmental degradation pathway of chloroplasts, such as *NYC1*, are instrumental in premature leaf senescence [11]. Additionally, the transcription factor OsERF101 directly interfaces with the promoter regions of *OsNAP* and *OsMYC2*, thereby activating genes implicated in chlorophyll degradation and JA-mediated leaf senescence [12]. The high light-induced leaf senescence in rice features the participation of the rice gene LRR-like1 *YELLOW AND PREMATURE DWARF 1* [3]. Moreover, the *OsNAP* gene encoding a plant-specific NAC transcriptional activator is an important senescence regulator. Altered *OsNAP* function significantly decelerates senescence whilst augmenting photosynthetic capacity, making it advantageous for high rice yields [13]. The rice gene *ES1* encodes a SCAR-like protein, which notably boosts the number of stomata per unit leaf area, thus precipitating increased water loss—and, consequently, senescence [14]. Another gene, *LPS1*, encodes the iron–sulfur subunit *SDH2-1* of succinate dehydrogenase and exerts an influence on leaf senescence and rice yield [15]. Senescence of plants can be induced by many factors, including reproductive growth, phytohormones and environmental cues; moreover, the senescence-associated genes regulated during early leaf senescence remain largely unknown.

Plant endogenous hormones also shape leaf senescence, making them popular research commodities with respect to understanding their regulatory mechanisms. As senescence sets in, plants tend to exhibit increasing levels of endogenous jasmonic acid and heightened expression of genes associated with jasmonic acid synthesis and signaling [1]. The *OsDOS* gene orchestrates plant senescence via a negative regulation of the signaling process of jasmonic acid [16]. Senescent leaves often witness an upsurge in the level of abscisic acid in vivo, which prompts the breakdown of chloroplasts [17]. Recent scientific advances have revealed that *ABI5* is a direct regulator of the *NAC* transcription factor gene *ORESARA1*’s expression [18]. Introduction of the promoter of the *SAG12* gene and the gene encoding isoprenyl transferase (a cytokinin synthesis gene) to a plant can delay its developmental senescence process [19]. Moreover, genes that induce cytokinin synthesis can effectively attenuate drought-triggered senescence [17]. Gibberellins, on one hand, stave off leaf senescence but can also provoke senescence by inducing reproductive growth. Evidence of this was reported in a study demonstrating that *AtWRKY75* positively regulates age-related leaf senescence via the gibberellin pathway [20]. Auxin can also modulate leaf senescence levels via auxin effectors. An interaction between *Stg3 (ZmATG18b)* and *Stg7 (ZmGH3.8)*, on the other hand, can regulate leaf senescence timing in maize [21].

Leaf tip withering in rice is a significant leaf senescence problem, manifesting as browning and necrosis at the leaf tips, potentially leading to reduced photosynthetic efficiency and lower grain yield. Leaf tip withering can result from both genetic mutations and environmental stresses [4]. Genetic factors include mutations in the NAD salvage pathway, such as *lts1,* encoding *OsNaPRT1* (*O. sativa* NaPRTase 1) in rice. A point mutation *lts1* in the NAD-compensated synthesis pathway—specifically, the gene nicotinic acid phosphoribosyltransferase (*OsNaPRT1*)—spawns a unique phenotype in rice characterized by dwarfism and leaf tip wilting. The leaves of these mutants exhibit premature senescence compared to their wild-type counterparts [4]. *OsCKI1* (casein kinase I1) plays a crucial regulatory role in a number of important physiological processes; loss of *OsCKI1* results in excess starch in leaves and a distinct leaf tip wilting phenotype, as well as high ROS and MDA levels, low chlorophyll content and protective enzyme activities [22]. The *OsPG1* gene encodes a polygalacturonase involved in pectin degradation and cell wall integrity, playing a vital role in leaf development and stress response. Mutations in *OsPG1* have been linked to necrosis appearing in leaf tips at the tillering stage [23]. Despite in-depth study of these mutants, the linkage between development and leaf tip senescence is still poorly understood in rice.

Repetitive elements (REs), such as transposable elements (TEs) and satellites, comprise much of the rice genome. More than 60% of lncRNAs are associated with transposable elements [24]. TEs and (peri)centromeric satellite DNA, which may contribute to leaf senescence via microRNA *SAG12* (senescence-associated genes), is repressed by the microRNA *miR164* via cleavage of *ORE1* [25]. The overexpression of another microRNA, *miR319*, causes a stay-green phenotype mRNA [26]. *miR390* triggers the production of the trans-acting siRNA *TAS3*, which results in the mRNA degradation of the auxin response factors *ARF2*, *ARF3* and *ARF4* [27]. In-depth study of these TEs or microRNA-associated mutants will help discover the linkage between development and senescence in rice. These research endeavors will furnish the theoretical foundation required for the regulatory control and utility of leaf senescence. In our study, in order to address this knowledge gap, our study focuses on a rice mutant exhibiting dwarfism and wilted leaf tips following ethyl methanesulfonate (EMS) treatment. We conducted comprehensive investigations into the phenotype, photosynthetic characteristics, chloroplast anatomy, reactive oxygen species (ROS) activity and the expression profiles of senescence-associated genes. 

## 2. Results

### 2.1. The lts2 Mutants Exhibited Dwarfism and Withered Leaf Tips throughout the Plant 

Only one wilted leaf tip material was screened from the 1500 M2 population of the Indica Group rice variety ‘Changchungu’ induced by ethyl methanesulfonate (EMS) mutagenesis and designated *lts2*. The *lts2* mutants exhibited both withered leaf tip phenotypes from seedling till harvesting (Figure 1a). Alongside withered leaf tips, the *lts2* mutants also presented narrower and shorter leaves compared to the wild type (Figure 1a,b). Additionally, vein sizes were smaller in *lts2* mutants, as confirmed by cross-sections (Figure 1b). Relative to the wild type (WT), the *lts2* mutants had a reduction in various growth parameters: tillering (76%), plant height (64%), grains per panicle (24%), seed setting rate (27%) and the number of primary (71%) and secondary (21%) branches (Figure 1c–h).

### 2.2. Analysis of Photosynthetic Rate, Pigment Content and Gene Expression

Chl a, Chl b and carotenoid (Car) content were lower in *lts2* mutants, which aligns with the visibly observed leaf senescence (1.09 mg/g FW, 0.38 mg/g FW and 0.30 mg/g FW, respectively) compared to the wild type (1.75 mg/g FW, 0.56 mg/g FW and 0.42 mg/g FW, respectively) in the upper leaves at the tillering stage (Figure 2a). Consequently, the photosynthetic rates were significantly reduced in *lts2* mutants (12.8 μmol CO_2_ m^−2^ s^−1^) relative to the wild type (22.3 μmol CO_2_ m^−2^ s^−1^) (Figure 2b). These results imply the involvement of an internal factor in leaf senescence affecting overall plant growth in *lts2* mutants. Gene expression analysis for those associated with chlorophyll synthesis and chloroplast development revealed significantly lower levels of *CAO, PORB, CHLH, CLHD, CHLL, DVR, OsHEMB, Lxhp2, psbA, RPS15, V1* and *V2* in *lts2* plants compared to the wild type (*p* < 0.01, Student’s *t*-test; Figure 2c,d), whereas *RPOC1* and *RPOC2* exhibited higher expression in *lts2* plants (Figure 2c,d).

### 2.3. lts2 Affects the Leaf Stomata

The early senescence mutant rice *ES1* displays a higher stomatal density than wild-type rice [14]. To ascertain if *lts2* shared this phenotype, a comparative analysis of *lts2* and wild-type leaves using scanning electron microscopy was performed. No significant difference in the stomatal density or length was noted (Figure 3a,b,d); however, the stomatal width was approximately 30% greater in *lts2* plants (Figure 3c,e).

### 2.4. Analysis of Chloroplast Ultrastructure

In pursuit of the causal factors underlying low chlorophyll levels in *lts2* during senescence, we investigated mesophyll cell chloroplast ultrastructure through transmission electron microscopy (TEM). The *lts2* mutant cells had significantly more osmiophilic granules compared to the wild type (Figure 4a). An immunoblot analysis evaluating four LHCI proteins—LHCA1, LHCA2, LHCA3 and LHCA4—also confirmed a pronounced reduction in these proteins in *lts2* mutants (Figure 4b), suggesting abnormal chloroplast development in *lts2*.

### 2.5. lts2 Plants Exhibit Symptoms of Premature Cell Death

Reactive oxygen species (ROS) play an important role in cell injury and cell death, and the accumulation of ROS is a trigger for premature senescence. Therefore, we have used the NBT and DAB staining techniques to identify ROS in the upper leaves (at the seedling stage) of both the WT and the mutant plants. We observed significant DAB staining (Figure 5a) and extensive NBT staining (Figure 5b) in *lts2* plants, indicating that the accumulation of ROS in these plants may lead to senescence of rice leaves, in contrast to WT leaves. We measured the levels of senescence-related substances such as hydrogen peroxide (H_2_O_2_)-derived ROS and the lipid oxidation by-product malondialdehyde (MDA). The H_2_O_2_ content of *lts2* was higher than that of the WT (16.3 μmol/g and 9.08 μmol/g, respectively) (Figure 5c), and the ORF concentration and MDA content of *lts2* leaves was higher than that of the wild type (0.77 and 0.49 nmol/g.min, 10.96 and 16.94 nmol/g, respectively) (Figure 5d,e), indicating the accumulation of ROS during premature leaf senescence in the *lts2* mutant. ROS-scavenging enzymes, including catalase (CAT), superoxide dismutase (SOD) and peroxidase (POD), have critical regulatory functions in plant senescence. The activity of these enzymes was quantified. In contrast to the wild type, CAT activity was significantly lower in *lts2* (1051 U/g fresh weight and 663 U/g fresh weight, respectively) (*p* < 0.05, Student’s *t*-test; Figure 5f,g). Conversely, POD activity was significantly higher in *lts2* leaves compared to wild-type leaves (2242 U/g fresh weight and 2607 U/g fresh weight, respectively) (Figure 5h). Since ROS scavenging systems play an important role in detoxifying ROS, we identified the expression levels of genes associated with ROS scavenging.The levels of *SODA, SODB, CATA, CATB, CATC, POD, AXO1A* and *AXO1B* were significantly higher in *lts2* plants compared to the wild type (*p* < 0.01, Student’s *t*-test; Figure 5i).

The mutants exhibited early onset of leaf senescence in the rice field, indicating that the *lts2* mutation may prompt cell death in rice. To confirm senescence in *lts2* plants, the expression levels of specific genes associated with chlorophyll degradation, the chlorophyll degradation genes (CDGs) (including *NCYC1, NOL, NYC3, NYC4, PAO, SGR, RCCR1* and *RCCR2*) and other senescence-associated genes (SAGs) (*Osh36, Osl57* and *Osl85*) were assessed using reverse transcription quantitative PCR (RT-qPCR). Expression of *SGR, RCCR1, RCCR2* and *Osl85* was notably higher in *lts2* plants compared to the WT (Figure 6a,b).

### 2.6. Map-Based Cloning and Candidate Analysis of the lts2 Gene

The *lts2* mutant was crossed with the Japonica Group cultivar NPB to generate F_2_ populations for the investigation of the molecular mechanism behind the phenotype. All F_1_ plants exhibited a green leaf phenotype akin to that of the WT. Among the 2126 F_2_ plants derived from the *lts2* mutant line and Nipponbare crosses, 1566 showcased the normal green phenotype, while 560 showed leaf senescence resembling that of *lts2*. This segregation pattern aligns with an approximately 3:1 ratio according to the *X^2^* test (*X^2^* = 2.04, *p* = 15.3%), suggesting that the *lts2* trait is governed by a single recessive nuclear gene. Using individuals exhibiting the *lts2* phenotype from the F2 population, primary mapping localized the *lts2* locus on chromosome 1, between markers B1-12 and B1-16 (Figure 7a). Subsequent fine mapping with additional SSR markers delimited the *lts2* to a 212 kb region between markers Q1 and L12, where occurrences of three and two recombinants were found, respectively. However, it is difficult to design new SSRs or indel molecular marker primers within this interval. Therefore, we re-checked the localization interval using the bulk segregant analysis (BSA) experiment. Bulk segregant analysis (BSA) further identified a large effect single-nucleotide polymorphism (SNP) within this region (Figure 7b,c) and a single nucleotide substitution (C-T) in the exon of LOC_Os01g35860 in *lts2* plants. In addition, we also performed one-generation sequencing of other candidate genes (LOC_Os01g34480, LOC_Os01g34660, LOC_Os01g34930, LOC_Os01g35990, LOC_Os01g36294, LOC_Os01g36350, LOC_Os01g36600, LOC_Os01g36840, LOC_Os01g37050, LOC_Os01g37280, LOC_Os01g37837, LOC_Os01g38990, LOC_Os01g39270) that may be involved in premature ageing. However, no mutations were found. We have therefore tentatively identified LOC_Os01g35860 as a candidate gene and assert that further studies are needed to identify the candidate *lts2* gene responsible for the premature senescence leaf phenotype.

### 2.7. RNA Sequencing Analysis of lts2 Mutants

RNA sequencing was conducted to elucidate the *lts2* mutation’s impact on gene expression. Leaf tip samples were collected from both *lts2* and WT plants at the seedling stage and were analyzed in triplicate. The reproducibility between biological replicates was exemplified by the consistent expression levels in box plots comparing WT and *lts2* samples (Figure 8a). In total, 23,855 genes were identified, with 1403 genes being up-regulated and 966 genes down-regulated in *lts2* compared to WT (Figure 8b,c). To further elucidate the regulation mechanism of leaf tip senescence, the differentially expressed genes related to chlorophyll degradation, senescence-associated genes, chlorophyll synthesis and chloroplast development genes and ROS detoxification genes were comprehensively analyzed from 2369 differentially expressed genes (DEGs). In particular, four chlorophyll degradation genes *SDG, OsRCCR1, OsRCCR2, OsRCCR3* were found to be up-regulated in *lts2*. In the chlorophyll synthesis and chloroplast development pathway, three genes were found to be down-regulated in *lts2*. A senescence-associated gene, *Osh36*, also shows high expression in *lts2*. Since ROS scavenging systems play an important role in detoxifying ROS, we identified five genes (*APX2, SODB, CATA, AXO1A* and *AXO1B*) associated with ROS scavenging that show significantly higher expression in *lts2* plants compared to the wild type (Appendix A). Taken together, the transcription or abundance of these genes increased and decreased, revealing changes in chlorophyll metabolism and ROS homeostasis in *lts2* leaves, which was basically consistent with the RT-qPCR detection results.

We subsequently performed a GO enrichment analysis of these 2369 DEGs. Gene ontology and Kyoto Encyclopedia of Genes and Genomes (KEGG) pathway analyses revealed significant alterations in genes involved in plant hormone signaling, such as *CYP714B2(Os03g0332100), OsARF9(Os04g0442000), OsIAA25(Os08g0109400), OsIAA21(Os06g0335500),* and *OsARF6a(Os02g0164900);* and in the MAPK signaling pathway, such as *OsMAPK2 (Os08g0157000), OsMPKK10.2 (Os03g0225100), OsMAPK6 (Os06g0154500), OsMAPK4 (Os06g0699400),* and *OsMAPK33 (Os02g0148100)* in *lts2* mutants (Appendix A). These results suggested that the mutation of *lts2* might affect the expression of plant hormone—and MAPK signaling—related genes in an indirect manner.

## 3. Discussion

Leaf senescence is meticulously governed by a genetic program. Various genes implicated in leaf senescence have been cloned and functionally characterized in rice and other plants [1,3,12,15,20,28]. For instance, *YPD1* encodes an LRR-like1 protein and is involved in high light-induced leaf senescence as well as chloroplast development in rice [3]. Besides genetic factors, abiotic stresses such as low nitrogen, drought, salinity, darkness, hypoxia, extreme temperatures, elevated sugar levels and UV radiation have also been shown to trigger premature leaf senescence and inhibit plant growth [29,30], suggesting that leaf senescence and plant growth might be co-regulated by overlapping genetic elements. Despite the identification of numerous genes associated with leaf senescence in rice, the role of transposons in this process remains elusive. A 3-bp deletion in the *WLS5* gene, whose product has an undetermined function, led to weakened growth and premature leaf senescence in rice [1]. Similarly, a point mutation in the NAD biosynthesis pathway gene *OsNaPRT1* led to plants with dwarf features and leaves exhibiting premature senescence when compared to wild-type plants [4]. NAD and its derivative NADP are crucial metabolites for redox reactions in living organisms and form a basis for almost every metabolic pathway in the cell [4]. In this study, we characterized a novel leaf-senescing mutant, *lts2*, which may encode a transposable element protein. Notably, *lts2* mutant plants demonstrated signs of leaf senescence throughout their development (Figure 5) and were also characterized by dwarf stature and reduced organ size (Figure 1a,b). It was observed that a lower vein count accounted for the abnormal growth patterns in *lts2* when compared to the wild type. Like *lts1*, several genes encoding NAD(P)-binding domain-containing proteins show increased or decreased transcription or abundance in the transcriptomes (such as Os10g0417600, Os09g0491820, Os10g0113900, Os06g0709000, Os02g0826200, Os09g0127300, Os02g0701900, Os03g0818200, Os05g0186300, Os04g0630100) in *lts2*. The Os10g0417600 knockout reduced photosynthetic efficiency as represented by the maximum carboxylation rate of Rubisco (V_max_), the maximum electron transport rate (J_max_) and the chlorophyll fluorescence parameter Φ_PSII_. Therefore, we speculate that the senescence of the leaf tip in *lts2* may have the same pathway as that in *lts1*. Leaf senescence is typically correlated with diminished chlorophyll content and heightened levels of various ROS [1,3]. In the *lts2* mutants, senescence positively correlated with both lower chlorophyll levels and a surge in ROS accumulation. Our investigations exposed a decrease in chlorophyll levels and photosynthetic rate in the *lts2* mutant leaves (Figure 2a,b). Moreover, genes responsible for chlorophyll biosynthesis and chloroplast development were found to be markedly downregulated in *lts2* plants compared to wild-type ones (Figure 2c,d). RNA sequencing has also revealed that a number of genes related to chlorophyll metabolism and ROS homeostasis are significantly altered in lts2. Therefore, we speculate that degradation of chlorophyll and chloroplasts plays an important role in leaf senescence in *lts2* plants. Two pathways, PSI and PSII, are responsible for electron transfer during photosynthesis. The role of Photosystem I (PSI), enclosed by four transmembrane light-harvesting complex I (LHCI) subunits, namely Lhca1–Lhca4, resulting in a PSI-LHCI supercomplex, is to collect light energy and subsequently transmit it to the PSI core to activate both charge separation and electron transfer reactions [31]. The antenna of photosystem I (PSI) Chl a/b-binding proteins (Lhca1–4) displayed distinctly decreased levels in *lts2* compared to the wild type (Figure 4b). Therefore, we speculate that the photosynthesis was affected (Figure 2b), possibly because PSI inhibited the protein expression levels of LHCA1, LHCA2, LHCA3 and LHCA4 in *lts2.* These results demonstrated that loss of function of *lts2* affects the expression of genes associated with photosynthesis, Chl biosynthesis, chloroplast development and leaf senescence, which further provided evidence that *lts2* participates in chloroplast development, function or senescence. Thus, we surmise that the degradation of chlorophyll and deterioration of chloroplasts are pivotal factors in the senescence of *lts2* leaves. 

The steady-state levels of ROS are determined by the balance between the mechanisms of ROS clearance and of ROS production [32]. Thylakoid membranes, alongside other cellular structures, can endure oxidative damage due to ROS [1,27]. The ROS H_2_O_2_ is produced endogenously by plant pathways, and excessive H_2_O_2_ production is characteristic of PCD. Our data showed that a large amount of H_2_O_2_ accumulated in *lts2* leaves, as confirmed by several histochemical staining experiments and determination of H_2_O_2_ levels (Figure 5c). Furthermore, our analysis revealed that *lts2* leaves experienced a significant elevation in oxygen-free radicals (OFRs), a category of ROS, compared to wild-type leaves (Figure 5d), which suggested that higher cellular ROS levels were present in the mutant leaves. Based on GO analysis, many genes related to the GO terms ‘kinase activity’, ‘oxidoreductase activity’ and ‘oxidation-reduction process’ were found to be up- or down-regulated in the *lts2* mutant, suggesting that the content of ROS in *lts2* leaves may have changed. Taken together, these results suggest that the reduction in the activity of these three key enzymes for the removal of excess active oxygen in plants resulted in an inability to remove ROS from cells in a timely manner, leading to the accumulation of large amounts of ROS, which ultimately led to cell death in the *lts2* mutant. These findings underscore the relationship between leaf senescence in *lts2* plants and ROS.

Phytohormones play important regulatory roles in promoting or delaying leaf senescence. In our study, many DEGs such as *CYP714B2*(Os03g0332100), *OsARF9*(Os04g0442000), *OsIAA25*(Os08g0109400), *OsIAA21*(Os06g0335500) and *OsARF6a*(Os02g0164900) had lower expression in *lts2* (Appendix A). ARF2 has been reported to be a negative regulator of auxin responses involved in the timing of senescence and flowering. These altered levels of IAA-related genes in *lts2* mutants may alter the levels of endogenous IAA hormones, contributing to its leaf senescence phenotype.

We honed in on the *lts2* locus to a 212 kb interval on the long arm of chromosome 1 through a combination of map-based cloning and bulked segregant analysis (BSA). Currently, there is no known link between these genes and premature leaf senescence in rice. Upon comparing the sequences of these genes in wild-type and *lts2* plants, only a single nucleotide substitution was identified within the coding region of LOC_Os01g35860, which encodes a protein of unknown function. Although the transposon’s role in the mechanism of senescence is not fully understood and warrants further investigation, the *lts2* mutant presents an ideal subject for such analysis. Several miRNA or microRNA, such as *miR164*, *miR319* and *miR390*, have been shown to lead to changes in the expression of genes associated with senescence. Plant long non-coding RNAs (lncRNAs) are widely accepted to play crucial roles during diverse biological processes. More than 60% lncRNAs were associated with transposable elements, especially TIR/Mutator and Helitron DNA transposons families. The alterations of LOC_Os01g35860 in *lts2* plants may cause diverse biological processes which including wilted leaf tips. Previous studies have shown that three CDGs (*SGR, RCCR1, RCCR2*) and two SAGs (*OsI57* and *OsI85*) are highly expressed in senescing tissues of rice and are involved in the aging process [1,3,4]. In the present study, the up-regulation of these CDGs and SAGs was detected in the flag leaves of *lts2* at the tillering stage (Figure 6a,b). These results indicated that *lts2* accelerated leaf senescence in a SAG- and CDG-dependent manner.

Consequently, it is hypothesized that *lts2* may partake in complex growth regulatory networks within rice, influencing chloroplast development and programmed cell death. To tease apart *lts2*’s direct role from the indirect effects arising due to the *lts2* mutation, further studies are essential. Moreover, identifying proteins that interact with *lts2* should be a focus of future research to shed light on the molecular mechanisms in which *lts2* partakes.

## 4. Materials and Methods

### 4.1. Plant Materials

The *lts2* mutant was derived from the indica group variety ‘Changchungu’ following treatment with ethyl methanesulfonate (EMS). The WT seeds were treated by complete submergence at 0.4% EMS concentration for 12 h. After mutagenic treatment, the seeds were rinsed with 5% Na_2_S_2_O_3_ solution to terminate the reaction, and then the seeds were rinsed with running water for 2 h. The seeds were then sown in the field. The mutant traits were consistently observed through several generations of self-crossing and in numerous field observations, indicating stable inheritance.

### 4.2. Paraffin Sectioning and Microscopic Analysis

Leaves of wild-type and *lts2* plants were harvested at the tillering stage and fixed in a 50% ethanol, 0.9 M glacial acetic acid and 3.7% formaldehyde solution overnight at 4 °C. The samples underwent dehydration in a graded ethanol series followed by xylene infiltration and were then embedded in paraffin (Sigma, Tampa, FL, USA). Specimens were sectioned at 8 mm using a Leica RM2245 microtome(Leica, Wetzlar, Germany), transferred to poly-L-lysine-coated glass slides and deparaffinized. Staining with 1% safranin (Amresco, DE, USA) and 1% Fast Green (Amresco, DE, USA) preceded a final dehydration through a graded ethanol series and xylene infiltration. Mounted sections were examined under a Nikon SMZ1500 microscope (Nikon, Tokyo, Japan).

### 4.3. Measurement of Pigment Content and Photosynthetic Rate

Fresh leaf samples (0.05 g) from both wild-type and *lts2* plants were homogenized and incubated with 5 mL of 80% acetone for 48 h in the dark. Absorbance of the acetone extracts was read using a DU800 ultraviolet spectrophotometer (BECKMAN, Fullerton, CA, USA) at 470 nm, 645 nm and 663 nm for the quantification of total chlorophyll (Chl), chlorophyll a (Chl a), chlorophyll b (Chl b) and carotenoids (Car) using formulas established by Arnon and Wellburn. This protocol was replicated with ten biological samples per condition. Photosynthetic rates were measured 65 days post-sowing at 10:30 a.m. on clear days using a LI-6400 portable photosynthesis system (LICOR, Lincoln, NE, USA) for both plant types, with fifteen replicates each.

### 4.4. Transmission Electron Microscopy (TEM)

TEM analysis adhered to the procedures established by Chen et al. [3]. Fresh leaf samples from wild-type and lts2 plants were initially immersed in 2.5% glutaraldehyde within phosphate-buffered saline (PBS, 137 mM NaCl, 2.7 mM KCl, 8 mM Na2HPO4+ and 2 mM KH2PO4, pH7.4)) for a minimum of 4 h, then washed in PBS three times. After post-fixation in 1% (*w/v*) OsO4, samples were dehydrated, embedded using a Spurr resin kit (Sigma) and ultra-thin sectioned at 70 nm using a Leica EM UC7 ultramicrotome(Leica, Wetzlar, Germany). Sections were contrasted with uranyl acetate and lead citrate before being examined with a Hitachi H-7650 electron microscope (Hitachi, Tokyo, Japan).

### 4.5. Histochemical Staining

Methods outlined by Zhao and Chen [1,3] were employed. Superoxide anion accumulation was detected using nitro blue tetrazolium (NBT, 0.5 mg/mL), and hydrogen peroxide was visualized with 3,3′-diaminobenzidine (DAB, 1 mg/mL). Manual leaf staining with trypan blue allowed for assessment of cell death. Additionally, for wild-type and *lts2* leaves, enzyme-linked quantifications of hydrogen peroxide (H_2_O_2_), oxygen free radicals (ORF), malondialdehyde (MDA) and antioxidant enzymes catalase (CAT), superoxide dismutase (SOD) and peroxidase (POD) were conducted using commercial assay kits (Suzhou Keming Biotechnology Co., Ltd., Suzhou, China).

### 4.6. Map-Based Cloning

A genetic analysis was conducted on the F_1_ and F_2_ populations that were generated from reciprocal crosses between *lts2* mutants and Nipponbare. The leaves of all F_1_ plants from crossbreeding was consistent with that of the wild type. Out of the 2126 F_2_. plants from the *lts2* mutant line and Nipponbare cross, 1566 displayed the wild-type phenotype and 560 displayed the mutant phenotype. The segregation ratio of the wild type to the mutant phenotype was approximately 3:1 raito according to *X*^2^ test (*X*^2^ = 2.04, *p* = 15.3%).

Over 560 F_2_ mutant individuals from a cross between *lts2* and ‘Nipponbare’ were subjected to genetic mapping. Genomic DNA was extracted using the CTAB method [33], and polymorphic simple sequence repeat (SSR) and sequence-tagged site (STS) markers specific to NPB/‘Changchungu’ were applied. Genomic fragments surrounding *LTS2* were amplified and sequenced using capillary electrophoresis for definitive alignment within an agarose gel matrix, provided by Hangzhou Tsingke Biological Engineering Technology and Service Co. Ltd., (Hangzhou, China).

### 4.7. Quantitative Real-Time PCR

Total RNA was isolated from fresh leaf samples of wild-type and *lts2* plants using TRIzol reagent, and first-strand cDNA was synthesized from 1 µg of RNA via ReverTra Ace qPCR RT Master Mix with gDNARemover. qPCR was conducted on an ABI PRISM 7900HT Sequence Detector (Applied Biosystems) according to the manufacturer’s instructions. Primers for RT-qPCR are listed in Appendix A. The rice ubiquitin gene was used as an internal control. Data are reported as mean ± SD of three biological replicates, and statistical significance was assessed using Student’s *t*-test.

### 4.8. RNA-seq Analysis

Total RNA was extracted from the seedlings of both wild-type and *lts2* plants. mRNA was isolated using oligo-dT beads and reverse-transcribed using random hexamer primers for cDNA synthesis. Library construction and sequencing were executed on an Illumina HiSeq 2000 platform (Novogene, Tianjin, China), generating 45 million and 40 million gene reads, respectively. DEGs were deemed significant if their log2 (fold change) surpassed 1 and q values were less than 0.05. Comprehensive annotations utilized GOseq for gene ontology, and pathway enrichment was explored using the Kyoto Encyclopedia of Genes and Genomes (KEGG) database.

### 4.9. Western Blot Analysis

Total proteins were extracted from wild-type and *lts2* rice at the seedling stage. Tissues were ground in liquid nitrogen and thawed in extraction buffer [50 mM Tris-HCl pH 7.5, 150 mM NaCl, 10% glycerol (*v/v*), 0.1% Nonidet P-40, 1 mM DTT, 1 mM PMSF and 1× complete protease inhibitor cocktail (Roche)] for 15 min on ice. The supernatant was collected by centrifugation at 12,000 g for 10 min at 4 °C. Total proteins were separated by SDS-PAGE gels (8%), transferred to polyvinylidene difluoride (PVDF) membranes (GE Healthcare, New York, NY, USA) and blotted with different primary antibodies. Antibodies against photosystem proteins (anti-Lhca1, anti-Lhca2, anti-Lhca3 and anti-Lhca4) were obtained from Agrisera (Beijing, China). Anti-β-actin was used as a control.

## 5. Conclusions

We have successfully cloned the *lts2* mutant, characterized by wilting leaf tips and localized it to a 212 kb interval on chromosome 1, which includes a single nucleotide substitution within a transposon. The *lts2* mutation detrimentally affects chloroplast development and triggers premature cell death, thus providing a valuable model for comprehensive senescence studies. Additionally, the narrow-leaf phenotype in the *lts2* mutant is an aspect of leaf senescence. The forthcoming molecular analysis of *lts2* should elucidate the significance of this gene in the senescence process.

## Figures and Tables

**Figure 1 ijms-25-07082-f001:**
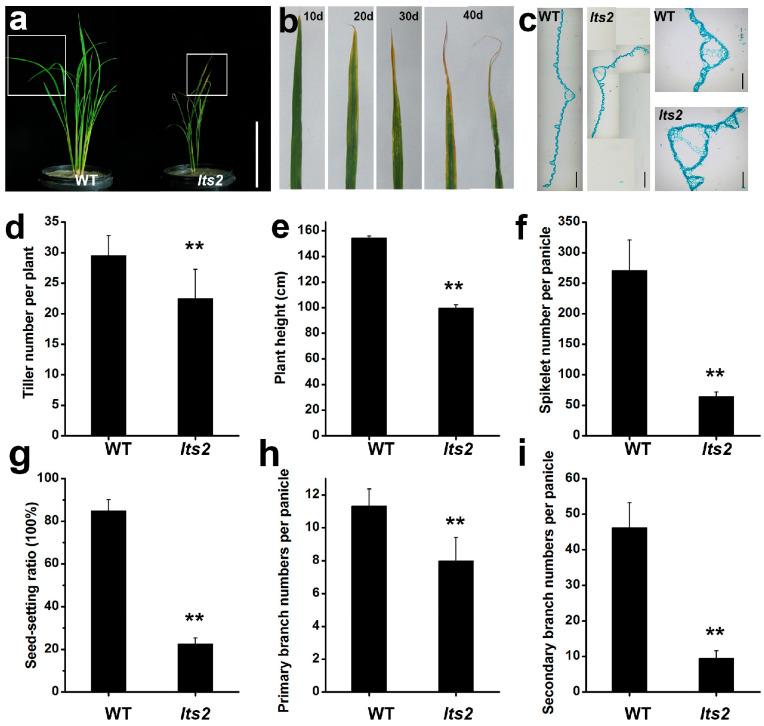
Phenotypic comparison of wild-type (WT) and *lts2* plants. (**a**) Morphological differences at the seedling stage, Bar = 20 cm. (**b**) Kinetic phenotypes of *lts2* leaves: left to right, at 30, 50, 70 and 90 d post-germination, respectively. (**c**) Cross sections of leaves. Bar = 500 μm, (**d**–**i**). Statistical analysis of tiller, plant height, grains per panicle, setting rate, primary branch and secondary branch between WT and *lts2* plants. Values are means ± SD of three biological replicates (** *p* < 0.01, Student’s *t*-test).

**Figure 2 ijms-25-07082-f002:**
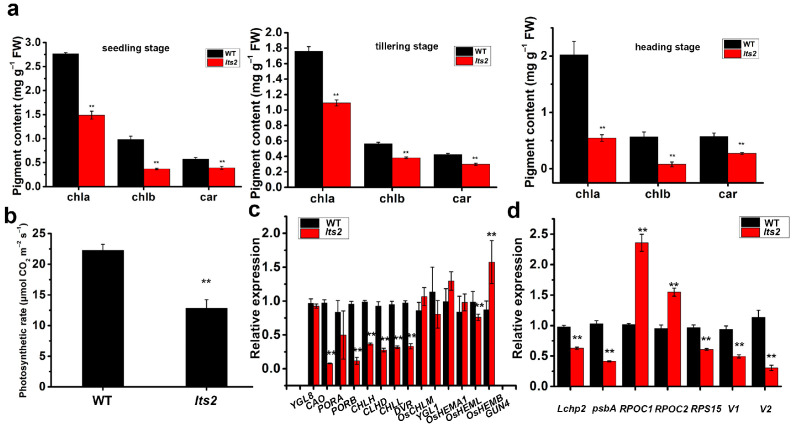
Comparison of leaf senescence indicators between wild-type and *lts2* plants. (**a**) Chlorophyll content in the upper leaves at seedling stage, tillering stage and heading stage. (**b**) Photosynthetic rate at tillering stage. (**c**,**d**) Gene expression linked to chlorophyll biosynthesis and chloroplast development at tillering stage. Error bars denote SD, n = 10 for (**a**), n = 15 for (**b**) and *p*-values from Student’s *t*-test (** *p* < 0.01).

**Figure 3 ijms-25-07082-f003:**
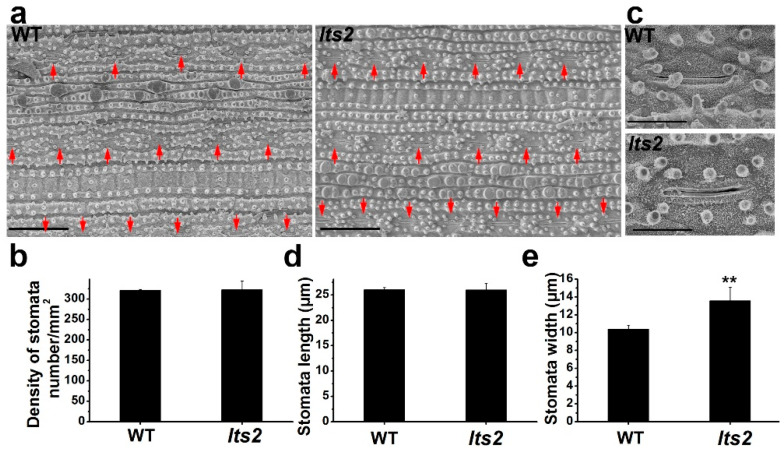
Observation of stomata in wild-type (WT) and *lts2* plants. (**a**) Stomatal density of WT and *lts2* leaves at tillering stage. Red arrows indicate the positions of stomata. Bars = 100 μm. (**b**) Statistical analysis of stomatal density. (**c**) Morphological characteristics of stomata of WT and *lts2* plants at tillering stage. Bars = 10 μm. (**d**,**e**) Statistical analysis of stomata length (**d**) and stomata width (**e**). Three independent replicates for stomatal density and 20 independent replicates for stomata size. ** *p* < 0.01 (Student’s *t*-test).

**Figure 4 ijms-25-07082-f004:**
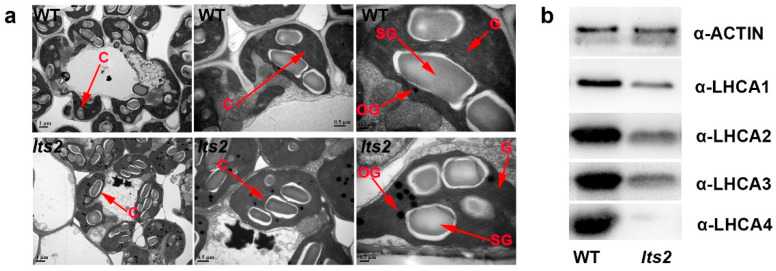
Chloroplast ultrastructure in WT and *lts2.* (**a**) Comparison of chloroplasts at tillering stage. C, chloroplast; OG, osmiophilic granule; SG, starch granule; G, grana thylakoid. (**b**) Immunoblots for LHCI proteins (LHCA1-4). Bars in (**a**) =1 μm and in (**b**) = 0.5 μm.

**Figure 5 ijms-25-07082-f005:**
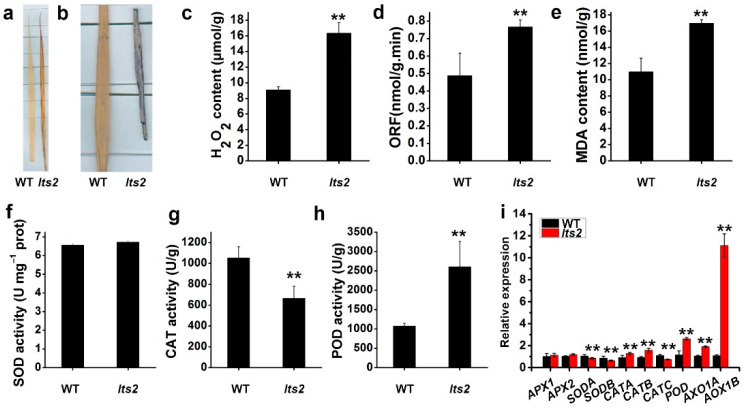
ROS accumulation in WT and *lts2* Plants. (**a**,**b**) DAB and NBT staining showing ROS localization at the seedling stage. (**c**–**h**) Quantification of H_2_O_2_ (**c**), ORF (**d**) and MDA (**e**) and activities of SOD (**f**), CAT (**g**) and POD (**h**) enzymes. Data represent mean ± SD from five replicates (** *p* < 0.01). (**i**), Differential expression of ROS detoxification genes, shown as mean ± SD from three replicates (** *p* < 0.01).

**Figure 6 ijms-25-07082-f006:**
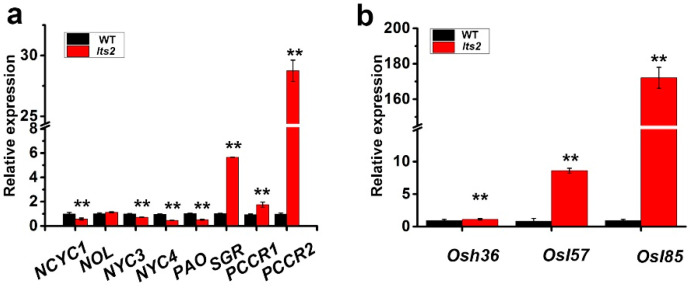
Cell death detection in WT and *lts2* plants. (**a**,**b**) Gene expression analysis of CDGs (**a**) and SAGs (**b**). Means ± SD of three independent replicates. ** *p* < 0.01 (Student’s *t*-test).

**Figure 7 ijms-25-07082-f007:**
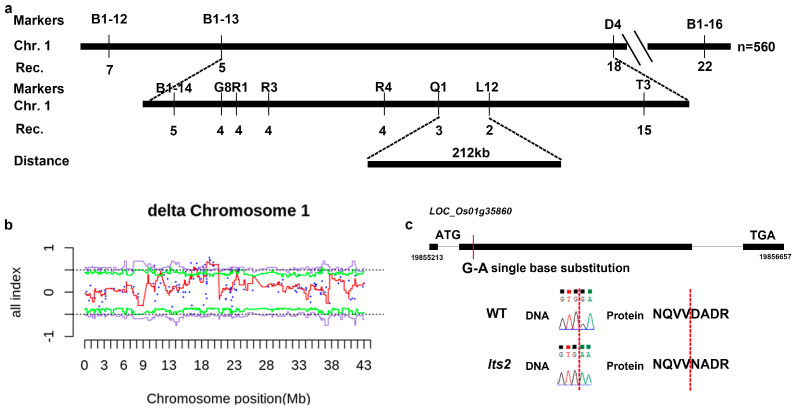
*lts2* candidate gene analysis. (**a**) Chromosomal localization of the *lts2* locus illustrated by marker data. Numbers represent recombinants identified. (**b**) Detection of a large-effect SNP through BSA on chromosome 1. (**c**) Detailed view of the structural and sequence variation in LOC_Os01g35860 between WT and *lts2*. Sequence_position from ATG-TGA is chr01:19855213..19856657.

**Figure 8 ijms-25-07082-f008:**
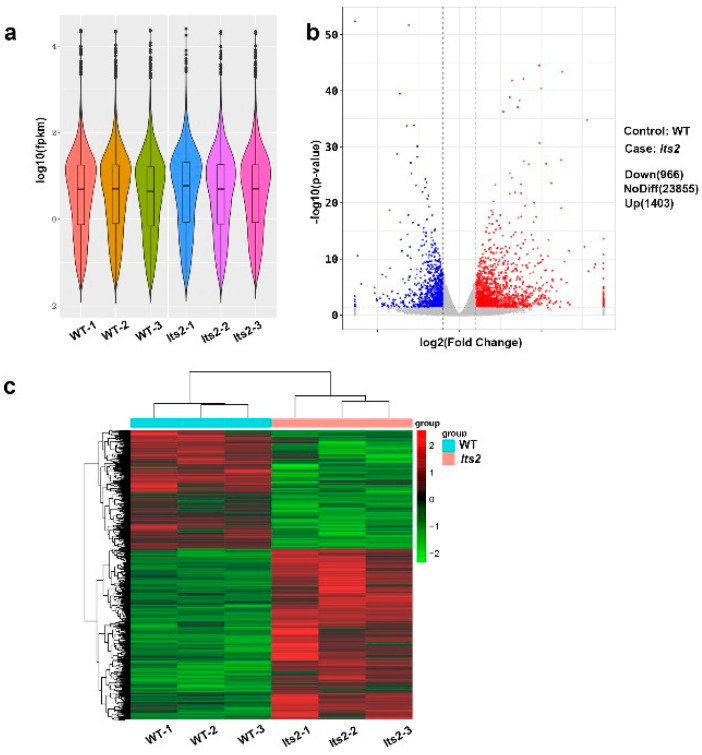
RNA-seq analysis in WT and *lts2*. (**a**) Comparison of transcript expression levels. (**b**) Volcano plot depicting gene expression changes, with up-regulated genes in red and down-regulated genes in blue. (**c**) Cluster analysis of differentially expressed genes.

## Data Availability

Data will be made available on request.

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
