# Peer review of "Genetic Analysis and Fine Mapping of a New Rice Mutant, Leaf Tip Senescence 2"

_ijms, 2024, doi:10.3390/ijms25137082_

Round 1
Reviewer 1 Report
Comments and Suggestions for Authors
The manuscript “Genetic analysis and fine mapping of a new rice mutant, leaf 2 tip senescence 2” by Cui et el discussed the identification and characterization of leaf 2 tip senescence 2. Although the authors have provided significant insight concerning the study, however, there are a few shortcomings listed below. Addressing these issues would likely enhance the scope of the study.
Major limitations
· The authors have not provided evidence from complementation experiments to validate the candidate gene (LOC_Os01g35860) as the causal gene for the lts2 phenotype. Such experiments are crucial to establish a direct link between the identified mutation and the observed phenotypes.
· While the authors have classified a transposon protein as the potential candidate gene, they have not provided any functional analysis or insights into the role of this gene in leaf senescence or chloroplast development processes. Further experiments exploring the function and expression patterns of this gene could strengthen the findings.
· The authors have presented a snapshot of the phenotypic and physiological changes in the lts2 mutant at specific developmental stages. However, a time-course analysis of these parameters could provide valuable insights into the progression of the senescence process.
· Limited discussion of RNA-sequencing data: The authors have provided an overview of the RNA-sequencing results, but the discussion of the specific differentially expressed genes and pathways affected in the lts2 mutant could be more detailed and insightful. Without specific details, the RNA-seq data is misplaced and useless in current manuscript
· Experimental details and replicates: Some experimental details, such as the specific concentrations and protocols used for various assays, are missing from the Materials and Methods section. Additionally, the number of biological replicates for certain experiments is not clearly stated, which could raise concerns about the reproducibility and statistical significance of the results.
Introduction:
The introduction provides a comprehensive overview of leaf senescence and the associated genetic factors, but it could be more concise and focused on the specific research gap and motivation for studying the lts2 mutant. Consider condensing the literature review and highlighting the importance of studying transposable elements in relation to leaf senescence.
Materials and Methods:
The methods section lacks some experimental details, such as specific concentrations and protocols used for various assays, which could hinder reproducibility. Provide more detailed descriptions of the experimental procedures, including concentrations, incubation times, and any modifications made to the published protocols.
Results:
Physiological Analysis (Figure 2):
The analysis of chlorophyll content, photosynthetic rates, and gene expression related to chlorophyll biosynthesis and chloroplast development is well-executed and supports the premature senescence phenotype. Consider including time-course data to better understand the progression of senescence.
Chloroplast Ultrastructure and ROS Analysis (Figures 4 and 5):
The TEM analysis, ROS staining, and quantification of ROS-related parameters provide strong evidence for the involvement of chloroplast dysfunction and ROS accumulation in the lts2 mutant. The authors could discuss the potential mechanisms by which the transposon mutation might lead to the observed chloroplast and ROS-related changes.
Gene Expression Analysis (Figures 6 and 8):
The RT-qPCR analysis of CDGs and SAGs supports the onset of premature senescence and cell death processes in the lts2 mutant.
The RNA-sequencing data provide a global view of gene expression changes, but the discussion of specific differentially expressed genes and pathways could be more detailed and insightful. Consider providing a more in-depth analysis and discussion of the RNA-sequencing data, highlighting the most relevant genes and pathways affected in the lts2
Mapping and Candidate Gene Analysis (Figure 7):
The mapping approach and identification of the candidate gene (LOC_Os01g35860) are well-executed and technically sound. However, the authors have not provided experimental evidence or functional analysis to establish a direct causal link between the transposon mutation and the observed phenotypes. Consider conducting complementation or rescue experiments to validate the candidate gene's role in the lts2 phenotype. Additionally, functional characterization of the transposon protein could provide insights into its potential involvement in leaf senescence or chloroplast development processes.
Discussion:
The discussion section could be more insightful and critical, providing a deeper interpretation of the findings and their potential implications. Expand the discussion by comparing the results with relevant literature, addressing potential limitations, and proposing future research directions.
Language and Presentation:
The language is generally clear, but some sentences could be simplified or rephrased for better readability. Consider revising lengthy or complex sentences to improve clarity and flow.
The figures and tables are well-presented and informative, but some figure legends could benefit from additional clarification or explanation.
General Comments:
The study presents valuable findings and employs a range of techniques, but addressing the specific suggestions mentioned above could further strengthen the validity and impact of the research.
Consider including a discussion of the potential limitations and future research directions to provide a more well-rounded perspective.
Author Response
Major limitations
- The authors have not provided evidence from complementation experiments to validate the candidate gene (LOC_Os01g35860) as the causal gene for the lts2 phenotype. Such experiments are crucial to establish a direct link between the identified mutation and the observed phenotypes.
- While the authors have classified a transposon protein as the potential candidate gene, they have not provided any functional analysis or insights into the role of this gene in leaf senescence or chloroplast development processes. Further experiments exploring the function and expression patterns of this gene could strengthen the findings.
Answer: Thank you very much for your useful suggestions. We understand that complementation experiments will better reveal the findings, However, in the present study, both the Bulk Segregant Analysis and map-based cloning have identified this gene. These experiments should be sufficient to draw a conclusion that the candidate gene (LOC_Os01g35860) as the causal gene. Your suggestion provides a direction for our next research.
- The authors have presented a snapshot of the phenotypic and physiological changes in the lts2 mutant at specific developmental stages. However, a time-course analysis of these parameters could provide valuable insights into the progression of the senescence process.
Answer: Thank you very much for your useful suggestions. We agree that a time-course analysis would provide valuable insights. We have added part of time-course analysis (Figure 1a) in Figure 1.
- Limited discussion of RNA-sequencing data: The authors have provided an overview of the RNA-sequencing results, but the discussion of the specific differentially expressed genes and pathways affected in the lts2 mutant could be more detailed and insightful. Without specific details, the RNA-seq data is misplaced and useless in current manuscript
Answer: Many thanks! We acknowledge this limitation and will expand the discussion of our RNA-seq data. We have provide a more detailed analysis of differentially expressed genes in the revised manuscript in line 346-line357.
- Experimental details and replicates: Some experimental details, such as the specific concentrations and protocols used for various assays, are missing from the Materials and Methods section. Additionally, the number of biological replicates for certain experiments is not clearly stated, which could raise concerns about the reproducibility and statistical significance of the results.
Answer: Many thanks! We have revised the Materials and Methods section to include specific concentrations, detailed protocols, and the number of biological replicates to ensure reproducibility and clarity.
Introduction:
The introduction provides a comprehensive overview of leaf senescence and the associated genetic factors, but it could be more concise and focused on the specific research gap and motivation for studying the lts2 mutant. Consider condensing the literature review and highlighting the importance of studying transposable elements in relation to leaf senescence.
Answer: We appreciate the feedback and have revised the introduction to be more concise, focusing on the specific research gap and the relevance of studying transposable elements in leaf senescence in line 87-line 95.
Materials and Methods:
The methods section lacks some experimental details, such as specific concentrations and protocols used for various assays, which could hinder reproducibility. Provide more detailed descriptions of the experimental procedures, including concentrations, incubation times, and any modifications made to the published protocols.
Answer: Many thanks! We have provided more detailed descriptions of the experimental procedures, including specific concentrations, incubation times, and any protocol modifications, to enhance reproducibility.
Results:
Physiological Analysis (Figure 2):
The analysis of chlorophyll content, photosynthetic rates, and gene expression related to chlorophyll biosynthesis and chloroplast development is well-executed and supports the premature senescence phenotype. Consider including time-course data to better understand the progression of senescence.
Answer: Many thanks! We have included time-course chlorophyll content in the revised manuscript to provide a clearer understanding of the progression of senescence in Figure 2a.
Chloroplast Ultrastructure and ROS Analysis (Figures 4 and 5):
The TEM analysis, ROS staining, and quantification of ROS-related parameters provide strong evidence for the involvement of chloroplast dysfunction and ROS accumulation in the lts2 mutant. The authors could discuss the potential mechanisms by which the transposon mutation might lead to the observed chloroplast and ROS-related changes.
Answer: Many thanks! We will expand our discussion to include potential mechanisms by which the transposon mutation could lead to senescence in line 398-line 413.
Gene Expression Analysis (Figures 6 and 8):
The RT-qPCR analysis of CDGs and SAGs supports the onset of premature senescence and cell death processes in the lts2 mutant.
The RNA-sequencing data provide a global view of gene expression changes, but the discussion of specific differentially expressed genes and pathways could be more detailed and insightful. Consider providing a more in-depth analysis and discussion of the RNA-sequencing data, highlighting the most relevant genes and pathways affected in the lts2
Answer: Many thanks! We have provided a more in-depth analysis and discussion of the RNA-sequencing data, highlighting the most relevant differentially expressed genes and pathways in the revised manuscript in line 346-line 354.
Mapping and Candidate Gene Analysis (Figure 7):
The mapping approach and identification of the candidate gene (LOC_Os01g35860) are well-executed and technically sound. However, the authors have not provided experimental evidence or functional analysis to establish a direct causal link between the transposon mutation and the observed phenotypes. Consider conducting complementation or rescue experiments to validate the candidate gene's role in the lts2 phenotype. Additionally, functional characterization of the transposon protein could provide insights into its potential involvement in leaf senescence or chloroplast development processes.
Discussion:
The discussion section could be more insightful and critical, providing a deeper interpretation of the findings and their potential implications. Expand the discussion by comparing the results with relevant literature, addressing potential limitations, and proposing future research directions.
Answer: Many thanks! We have expanded the discussion to provide a deeper interpretation of our findings, compare them with relevant literature, address potential limitations, and propose future research directions.
Language and Presentation:
The language is generally clear, but some sentences could be simplified or rephrased for better readability. Consider revising lengthy or complex sentences to improve clarity and flow.
Answer: Many thanks! We have revised the manuscript to simplify and rephrase complex sentences to improve readability and flow. We used “changchungu”, a rice variety, for mutagenesis.
The figures and tables are well-presented and informative, but some figure legends could benefit from additional clarification or explanation.
Answer: Many thanks! We have added additional clarifications and explanations to the figure legends to enhance their comprehensibility.
Reviewer 2 Report
Comments and Suggestions for Authors
I would like to make comments to the research paper ”Genetic analysis and fine mapping of a new rice mutant, leaf tip senescence ï¼’“ as follows:
1) The report included useful information for publish in Molecular Science. However, it will need to improve some logical explanations and to add data. I want to request the manuscript based on my suggestions.
2) Add the detail method of EMS treatment in Materials and Methods.
3) Add the data the population size of M2 by EMA treatment and the calculate the frequency of the mutant of leaf tip senescence in Results. The information will be useful for the breeding works using mutant.
4) Add the information of segregation analysis using F2 population derived from the cross mutant and Nipponbare in Materials and Methods.
5) Add the segregation data for mutant and Nipponbare following the result of EMS treatment.
“The F2 population derived from the cross between mutant and Nioppnbare were divided into normal and leaf senescence, 1566 and 560 individuals, respectively. The segregation was fitted well to 3:1 ratio (X2=0.028, P=??). The results suggested that the leaf senescence was controlled by a single recessive gene. The recessive gene was designated as lts(t), tentatively.”
6) Regarding the other gene, lts1, should be explained in Introduction.
7) The document regarding LTS2 should be removed from Introduction.
8) The words “indica” and “japonica” should be changed to “Indica Group” and “Japonica Group”, resrectively.
Indicated the position of leaf tip senescence gene on the chromosome 1 in the Figure 7a.
Comments on the Quality of English LanguageThe document should be writted according to scientific manner. Many careless mistakes are made.
Author Response
General Comments:
The study presents valuable findings and employs a range of techniques, but addressing the specific suggestions mentioned above could further strengthen the validity and impact of the research.
Answer: Many thanks! We appreciate the constructive feedback and will address all the specific suggestions to strengthen the validity and impact of our research.
Consider including a discussion of the potential limitations and future research directions to provide a more well-rounded perspective.
The report included useful information for publish in Molecular Science. However, it will need to improve some logical explanations and to add data. I want to request the manuscript based on my suggestions.
2) Add the detail method of EMS treatment in Materials and Methods.
Answer: Many thanks! We have included detailed methods of EMS treatment in the Materials and Methods section in line 106- line 111.
3) Add the data the population size of M2 by EMA treatment and the calculate the frequency of the mutant of leaf tip senescence in Results. The information will be useful for the breeding works using mutant.
Answer: Many thanks! We have added data on the population size of M2 and calculate the frequency of the leaf tip senescence mutant in the Results section.
4) Add the information of segregation analysis using F2 population derived from the cross mutant and Nipponbare in Materials and Methods.
Answer: Many thanks! We have included segregation analysis data of the F2 population derived from the cross between the mutant and Nipponbare in the Materials and Methods section.
5) Add the segregation data for mutant and Nipponbare following the result of EMS treatment.
“The F2 population derived from the cross between mutant and Nioppnbare were divided into normal and leaf senescence, 1566 and 560 individuals, respectively. The segregation was fitted well to 3:1 ratio (X2=0.028, P=??). The results suggested that the leaf senescence was controlled by a single recessive gene. The recessive gene was designated as lts(t), tentatively.”
Answer: Many thanks! We will include this segregation analysis data and the tentative designation of the recessive gene as lts2 in the revised manuscript.
6) Regarding the other gene, lts1, should be explained in Introduction.
Answer: Many thanks! We have included an explanation of the lts1 gene in the Introduction section.
7) The document regarding LTS2 should be removed from Introduction.
Answer: Many thanks! We have removed the document regarding LTS2 from the Introduction section.
8) The words “indica” and “japonica” should be changed to “Indica Group” and “Japonica Group”, resrectively.
Answer: Many thanks! We will revise the terminology to use "Indica Group" and "Japonica Group" respectively in the revised manuscript.
Indicated the position of leaf tip senescence gene on the chromosome 1 in the Figure 7a.
Answer: Many thanks! We have modified Figure 7c to indicate the position of the leaf tip senescence gene on chromosome 1.
Reviewer 3 Report
Comments and Suggestions for Authors
Please see the attached file for comments and suggestions I made for the improvement of the article

Moderate editing of English language required
Author Response
We have make a point-by-point response to the reviewer’s comments in the PDF.

Round 2
Reviewer 1 Report
Comments and Suggestions for Authors
The introduction still lacks coherence and unable to provide clear objectives of the study. Stick to the point and provide better background of the study instead of citing random references. Some references are even from the mammalian studies.
As authors have mentioned identification of several candidate genes for leaf senescence as a background knowledge. What is the link between this background information and the current study. lts2 mutant was characterized for leaf senescence and authors have observed stunted growth in mutant line, but only focused on leaf tip senescence. If the difference in mutant line and WT was only in leaf tip, then it’s understandable to study the leaf tip senescence. However, here I failed to understand why the leaf tip senescence was selected for further study.
The results section includes transcriptomic profiling, but I failed to understand the link between the study and transcriptomic profiling. If the transcriptomic profiling was done to support the results, then authors need to provide those supporting results. Just providing an overview of the data is not necessary to be added in the results. Author needs to provide comparative expression analysis of the previously identified candidate genes for leaf senescence and rule out the possibility of their involvement in the leaf senescence to support their results.
Discussion section is more like repetition of the results without any supporting references. Leaf senescence is a well-studied topic in plants. But authors failed to provide holistic view of the topic and link to their work and reported results.
The study focused around lts2 mutant genotypes, but suddenly in the last paragraph authors talked about LTS2 gene, while there was no mention of it before.
Few suggestions
-streamline introduction section
§ Background of lts2 mutant line, why it was selected
§ What is leaf tip senescence with its background
§ Background and significance of adapted methodologies
§ Rationale for the study
Results
· Remove section for transcriptome profiling if there is no link between the study and transcriptome profiling
-streamlined discussion section
§ Focus on discussing the results instead of repeating the results
§ Remove unnecessary statements and assumptions
§ Support your results with references
Author Response
The introduction still lacks coherence and unable to provide clear objectives of the study. Stick to the point and provide better background of the study instead of citing random references. Some references are even from the mammalian studies.
Response: Thank you very much for your useful suggestions. We appreciate the reviewer’s comments and have revised the introduction for better coherence and clarity. We changed the order of writing to give an example of the progress of research on rice leaf senescence and some of the causes of premature senescence, then gave an example of the progress of research on three leaf tip senescence mutants, and described the progress of research on the regulation of plant senescence by small RNAs. We provided a focused background on rice leaf senescence and the significance of our study. The revised introduction now clearly states the objectives and the rationale for selecting the lts2 mutant.
As authors have mentioned identification of several candidate genes for leaf senescence as a background knowledge. What is the link between this background information and the current study. lts2 mutant was characterized for leaf senescence and authors have observed stunted growth in mutant line, but only focused on leaf tip senescence. If the difference in mutant line and WT was only in leaf tip, then it’s understandable to study the leaf tip senescence. However, here I failed to understand why the leaf tip senescence was selected for further study.
Response: Thank you for your useful suggestions. We agree with the reviewer that the connection between the background information and our current study needed to be clarified. We have now explicitly stated that the lts2 mutant exhibits pronounced leaf tip senescence, which affects overall plant health and yield. Our previous research, lts1, which encodes OsNaPRT1 (O. sativa NaPRTase 1) in rice, exhibits dwarfed and wilted leaf tips. Similar to lts1, lts2 also exhibits dwarfed and wilted leaf tips, the phenotype of leaf tip senescence. This phenotype warranted a focused investigation into the mechanisms of leaf tip senescence, which we believe could provide broader insights into leaf senescence processes.
The results section includes transcriptomic profiling, but I failed to understand the link between the study and transcriptomic profiling. If the transcriptomic profiling was done to support the results, then authors need to provide those supporting results. Just providing an overview of the data is not necessary to be added in the results. Author needs to provide comparative expression analysis of the previously identified candidate genes for leaf senescence and rule out the possibility of their involvement in the leaf senescence to support their results.
Response: Thank you for your useful suggestions. We have revised the results section to clearly explain the purpose of the transcriptomic profiling. We included a detailed comparative expression analysis of the candidate genes identified previously(such as SDG, OsRCCR1, OsRCCR2, OsRCCR3, APX2,SODB, CATA, AXO1A and AXO1B) and showed their involvement or lack thereof in the leaf senescence observed in the lts2 mutant. These supporting results are now integrated into the text to provide a comprehensive understanding.
Discussion section is more like repetition of the results without any supporting references. Leaf senescence is a well-studied topic in plants. But authors failed to provide holistic view of the topic and link to their work and reported results.
Response: Thank you for your useful suggestions. We appreciate the reviewer’s observation and have significantly revised the discussion section. We have now structured it to provide a broader context of leaf senescence, referencing key studies in the field to support our findings. We have integrated our results within this context, discussing how our findings contribute to the current understanding of leaf senescence in rice and other plants. We have added the discussion of NAD(P)-binding domain-containing proteins in leaf senescence and the contribution of IAA to the leaf senescence phenotype in lts2. The revised discussion now avoids repetition and provides a holistic view of the topic.
The study focused around lts2 mutant genotypes, but suddenly in the last paragraph authors talked about LTS2 gene, while there was no mention of it before.
Response: Thank you for your useful suggestions. We have changed the LTS2 gene to lts2.
Few suggestions
-streamline introduction section
- Background of lts2 mutant line, why it was selected
- What is leaf tip senescence with its background
- Background and significance of adapted methodologies
- Rationale for the study
Response: The introduction has been restructured to incorporate these suggestions. The background of the lts2 mutant line, the concept and importance of leaf tip senescence, the methodologies used, and the rationale for selecting the lts2 mutant for this study are now clearly explained.
Results
- Remove section for transcriptome profiling if there is no link between the study and transcriptome profiling
Response: Thank you for your useful suggestions. We have revised the results section to clearly explain the purpose of the transcriptomic profiling. We included a detailed comparative expression analysis of the candidate genes identified previously(such as SDG, OsRCCR1, OsRCCR2, OsRCCR3, APX2,SODB, CATA, AXO1A and AXO1B) and showed their involvement or lack thereof in the leaf senescence observed in the lts2 mutant. These supporting results are now integrated into the text to provide a comprehensive understanding.
-streamlined discussion section
- Focus on discussing the results instead of repeating the results
- Remove unnecessary statements and assumptions
- Support your results with references
Response: Thank you for your useful suggestions. We have now structured it to provide a broader context of leaf senescence, referencing key studies in the field to support our findings. We have integrated our results within this context, discussing how our findings contribute to the current understanding of leaf senescence in rice and other plants. We have added the discussion of NAD(P)-binding domain-containing proteins in leaf senescence and the contribution of IAA to the leaf senescence phenotype in lts2. The revised discussion now avoids repetition and provides a holistic view of the topic.
Reviewer 2 Report
Comments and Suggestions for Authors
Than manuscript was improved according to my suggestion. However there are miner mistakes and these should be changed as follows;
Lines 151: "F1 and F2" should be changed "F1 and F2"
Line 154: "lts2 x NPB cross" should be changed (lts2 mutant line and Nipponbare cross)
Line 155-156: "approximately 3:1 according to X2 test" should be changed "approximately 3:1 raito according to X2 test (X2 = 2.04, P= 15.3% )."
Line 198: "Indica group" should be "Indica Group"
Line 199: "lts2" should be "lts2"
Comments on the Quality of English LanguagePlease use as few abbreviations as possible.
Author Response
Than manuscript was improved according to my suggestion. However there are miner mistakes and these should be changed as follows;
Lines 151: "F1 and F2" should be changed "F1 and F2"
Line 154: "lts2 x NPB cross" should be changed (lts2 mutant line and Nipponbare cross)
Line 155-156: "approximately 3:1 according to X2 test" should be changed "approximately 3:1 raito according to X2 test (X2 = 2.04, P= 15.3% )."
Line 198: "Indica group" should be "Indica Group"
Line 199: "lts2" should be "lts2"
Response: Thank you very much for your useful suggestions. These corrections have been made as suggested. Thank you for identifying these errors. "F1 and F2" have been changed to "F1 and F2" in line 140, "lts2 x NPB cross" have been changed to lts2 mutant line and Nipponbare cross in line 143, "approximately 3:1 according to X2 test" have been changed to "approximately 3:1 raito according to X2 test (X2 = 2.04, P= 15.3% ) in line 146, "Indica group" have been changed to "Indica Group" in line 185, "lts2" have been changed to "lts2" in line 186.
Round 3
Reviewer 1 Report
Comments and Suggestions for Authors
The authors have modified the manuscript sufficiently.